# Blue Titania: The Outcome of Defects, Crystalline-Disordered Core-Shell Structure, and Hydrophilicity Change

**DOI:** 10.3390/nano12091501

**Published:** 2022-04-28

**Authors:** Sergio A. Sabinas-Hernández, Justo Miguel Gracia Jiménez, Nicolás Rutilo Silva González, María P. Elizalde-González, Ulises Salazar-Kuri, Samuel Tehuacanero-Cuapa

**Affiliations:** 1Instituto de Física, Benemérita Universidad Autónoma de Puebla, Av. San Claudio y Blvd. 18 Sur, Col. San Manuel, Ciudad Universitaria, Apartado Postal J-48, Puebla 72570, Mexico; gracia@ifuap.buap.mx (J.M.G.J.); silva@ifuap.buap.mx (N.R.S.G.); usalazar@ifuap.buap.mx (U.S.-K.); 2Centro de Química, Instituto de Ciencias, Benemérita Universidad Autónoma de Puebla, Ciudad Universitaria, Edif. IC7, Puebla 72570, Mexico; maria.elizalde@correo.buap.mx; 3Instituto de Física, Universidad Nacional Autónoma de México, Circuito de la Investigación s/n, Ciudad Universitaria, Ciudad de México 04510, Mexico; samueltc@fisica.unam.mx

**Keywords:** blue titania, black TiO_2_, partially reduced titania, hydrophilicity of TiO_2_, near-infrared spectra of TiO_2_, crystalline-disordered core-shell structure, TiO_2_ P25

## Abstract

In this research, changes in several characteristics of partially reduced titania were studied. The reduction process used made it possible to gradually observe changes in the material depending on the amount of reducing agent used. We used NaBH_4_ to impregnate commercial TiO_2_ with isopropyl alcohol. Impregnated TiO_2_ nanoparticles were dried and thermally treated in a nitrogen flow to obtain blue titania samples. Thorough spectroscopic characterization showed that oxygen atoms from hydroxyl groups, as well as from the surface, and the lattice of TiO_2_ was consumed. This caused changes in the surface and even in the bulk of TiO_2_ when the amount of reducing agent used was increased. Structural, optical, superficial, and textural characteristics were studied using XRD, Raman, DRS UV-Vis-NIR, Mid-DRIFT, XPS, and nitrogen adsorption/desorption isotherms. A photocatalytic test of the degradation of methylene blue dye was performed. Among different effects on the mentioned characteristics, we found evidence of changes in the surface properties of the blue titania samples and their probable effect on the photocatalytic properties. The reduction process implied a preponderant decrease in the surface hydrophilicity of the reduced samples, an effect shown for the first time in this type of material.

## 1. Introduction

Semiconductor titania (TiO_2_) is a promising sunlight-harvesting material that has been widely investigated, and its capability has been experimentally proven in photocatalysis and other fields [1]. An ideal semiconductor photocatalyst should be able to catalyze reactions and be efficiently activated by sunlight. Furthermore, it must be chemically and thermally stable, have low toxicity, good photostability, and be able to be easily and cheaply produced [2]. TiO_2_ is close to being an ideal semiconductor according to the mentioned criteria. It shows almost all the listed properties; the only exception is that it poses a wide bandgap of energy (3.2 eV for anatase and 3.0 eV for rutile) [3] and therefore does not absorb visible light.

Various strategies have been used to tune TiO_2_’s optical properties to improve its visible light photoactivity: for example, the doping of titania with metallic and non-metallic ions, depositing metal nanoparticles, dye-sensitizing, etc. However, the absorption range of modified TiO_2_ does not extend to the infrared (IR) region [4,5]. Hydrogenated black TiO_2_ was reported for the first time by Chen et al. in 2011 [6], and their material was even able to absorb energy in the IR region. This partially reduced TiO_2_ was characterized by the incorporation of hydrogen doping or self-structural modifications which involved a self-doped Ti^3+^/oxygen vacancy [7]. The modified TiO_2_ also had a black, known as black titania, or colorful (i.e., yellow, green, blue, and brown) appearance [8,9]. After the discovery of black TiO_2_ nanoparticles, the synthesis of black TiO_2_ nanostructures has become a hot topic in contemporary environmental nanoscience [1].

According to the literature, crystalline defects in the bulk and on the surface play a crucial role in the optical, electronic, superficial, and, consequently, photocatalytic properties of reduced TiO_2_ nanomaterials. Examples of defects found in these types of materials are Ti^3+^ ions, oxygen vacancies, hydrogen doping [10], hydrogen-mediated oxygen vacancies [11], etc. It is even common to detect the presence of an amorphous phase or disordered layer around a crystalline TiO_2_ core [12]. In addition, the presence of Ti–OH and Ti–H species [1] is also important. All these defects and chemical species can contribute to a material’s ability to absorb in the visible up to the IR region of the electromagnetic spectrum and thereby change its photocatalytic performance.

The nature of the crystalline defects existing in reduced TiO_2_ nanomaterials depends on the method of synthesis used. Different synthetic pathways have been investigated for the synthesis of reduced TiO_2_, such as high/low-pressure hydrogenation treatment [6,13,14], chemical reduction [15,16,17,18], microwave irradiation [19], electrochemical reduction [20], ultrasonication [21], and plasma laser ablation [22,23]. The following different aspects have attracted attention: structural, optical, and photocatalytic properties. However, there exist some conflicting ideas related to the role of defects in the overall photoactivity of, black TiO_2_ and the location of these defects [24]. Additionally, it is important to mention that there has been little investigation of whether disordered or amorphous phases modify the surface properties of black TiO_2_, and this feature remains largely unaddressed [12].

In our research, commercial titanium dioxide, Aeroxide^®^ P25 of Ebonik Degussa, was used to prepare different reduced samples. TiO_2_ was modified by impregnation with different quantities of NaBH_4_ in an isopropyl alcohol medium and through thermal treatment in a nitrogen flow at a fixed temperature. Different characteristics of these materials were thoroughly studied using several techniques as a function of the reducing agent amount. We expect to introduce new insights related to the characterization of these types of materials and to the changes that TiO_2_ undergoes when it is reduced.

## 2. Materials and Methods

### 2.1. Materials and Reagents

The following reagents were used: Titanium (IV) oxide from Sigma-Aldrich (718467-100G, St. Louis, MO, USA); nanopowder Aeroxide^®^ P25 of Ebonik Degussa, known as Degussa P25, with an average size of 21 nm, purity ≥ 99.5%, and Lot. # MKBC4174; sodium borohydride NaBH_4_ from Aldrich (452874-25G, St. Louis, MO, USA), and Lot. # MKBZ2657V; isopropyl alcohol from J. T. Baker (9084-02, Phillipsburg, NJ, USA); Baker analyzed ACS; absolute ethanol, J. T. Baker; reactive Baker ACS (Cat. # 9000-03, USA), and Lot. # V32C55. For the photocatalytic test, deionized water with a resistivity of 18.2 MΩ cm and methylene blue dye from Sigma (M9140-25G, St. Louis, MO, USA) were used.

### 2.2. Preparation of Samples

A simple borosilicate glass distillation equipment was used for the preparation of the samples. The condenser was cooled with recirculated water at 20 °C. It was used all the time except when TiO_2_ was added. A certain amount of NaBH_4_ was dissolved in 40 mL of isopropyl alcohol in a 100 mL round-bottom flask with magnetic stirring and heating. At around 70 °C, 2 g of commercial titania was added into the round bottom flask and was maintained with magnetic stirring. After ~0.5 h, the temperature was raised to 75 °C for distillation (or evaporation) of isopropyl alcohol. Once the evaporation or distillation was completed, the modified TiO_2_ was dried at 80 °C on a stove for an additional time of 0.5 h. Then, the dried sample of TiO_2_ was put into a small test tube of 1.7 cm in inner diameter and 7.0 cm in length (thickness of the wall test tube was 0.3 cm). The test tube was placed horizontally into a glass tube and was thermally treated at 5 °C/min up to 400 °C. This temperature was maintained for 90 min in a nitrogen gas flow (50 mL/min); then, reduced titania was cooled to room temperature in the same flow. Interestingly, after the thermal treatment and the cooling of the samples, the powder in the test tube resulted in two well-divided sections along the tube: one that showed a blue color and the other, in the upper part of the test tube, that remained white. This white powder was removed from the test tube (approximately a depth of 3 mm along the tube) in all samples.

Finally, the reduced TiO_2_ was washed with ethanol and deionized water, ad dried for 2 h at 80 °C. By fixing all parameters, the reducing agent (NaBH_4_) quantity was varied to prepare different samples of reduced TiO_2_. Roughly, around 78% yield of reduced titania was obtained. Table 1 indicates the prepared samples, their codes, and the weight fraction of NaBH_4_, expressed in percentage, $w_{NaBH_{4}}=m_{NaBH_{4}}/\left(m_{NaBH_{4}}+m_{P25}\right)$, used in their preparation. A sample treated in the same way, without the addition of NaBH_4_, was used as reference.

### 2.3. Samples Characterization

Morphological and structural information of the samples was obtained using a transmission electron microscope (TEM) from Jeol (model JEM 2010F, Akishima, Tokyo, Japan) operating at 200 kV. The samples were deposited on a carbon film supported on a copper grid. To analyze the high-resolution transmission electron microscopy (HRTEM) images, the VESTA [25] and SIMULATEM [26] software was used. Additionally, an FE-SEM from Jeol (model JSM 7800F) was used to analyze the morphology of the reduced TiO_2_ samples. TEM and SEM micrographs were analyzed with the ImageJ software version 1.53e.

Samples’ X-ray diffraction (XRD) patterns were measured using an X-ray diffractometer from Panalytical (model Empyrean, Almelo, Netherlands) with Cu Kα radiation (λ = 1.5406 Å) and a Ni filter. The acceleration voltage and the applied currents were 40 kV and 35 mA, respectively. Data were collected with a step width of 0.17° (2θ) from 10° to 90° (2θ). The same amount of sample was measured in all cases. Changes in lattice constants, composition, and crystal sizes were investigated using Rietveld refinement. All refinements were implemented through the BGMN/Autoquan software version 4.2.22 [27] with the graphical user interface Profex version 4.2.1 [28]. To achieve the refinements, structural information of anatase and rutile was taken from Howard [29]. Additionally, quantification of anatase and rutile phases was estimated by the Spurr and Myers [30] equation, where the weight fraction of anatase is $f_{A}=1/\left(1+1.256\left(I_{R}/I_{A}\right)\right)$ and where $I_{A}$ and $I_{R}$ are the integrated intensities of the maximum peak (101) for the anatase and (110) for the rutile phases, respectively. Consequently, the weight fraction of rutile is $f_{R}=1-f_{A}$. Voigt functions were used to determine the $I_{A}$ and $I_{R}$ areas of the two reflections. Additionally, we carried out an X-ray diffraction analysis as proposed by Chen and Xia [31] to study the structural changes after reduction of TiO_2_. The construction of the crystal shape of the phases in the samples was performed using the WinXMorph software [32,33] version 1.54 together with the results of the Rietveld refinements. Furthermore, the crystal class 4/*m*2/*m*2/*m* [34] of anatase and rutile was considered, whose space groups are *I*4_1_/*amd* (No. 141) and *P*4_2_/*mnm* (No. 136), respectively.

Raman spectra were obtained using a spectrophotometer from Horiba Jobin Yvon (model LabRam HR) with a 632.8 nm (He-Ne) laser and a 50× objective on a Olympus BX41 microscope (both from Villeneuve-d’Ascq, France). The LabSpec5 software was used. A short time of acquisition and a neutral density filter D1 was used to avoid damage to the samples. The resolution was 0.5 cm^−1^. Raman line positions were corrected using a Si peak of 520.7 cm^−1^. The most intense peak of each spectrum was integrated within the range from ~83 to ~188 cm^−1^ (before the beginning of the following peak). The considered peak positions and widths are the averages of six measurements.

Diffuse reflectance spectroscopy in the ultraviolet/visible/near-infrared regions (DRS-UV-VIS-NIR) was used to obtain spectra of all samples. Measurements were carried out without dilution on a spectrometer coupled to an internal diffuse reflectance accessory, both from Agilent-Varian (model Cary 5000, Mulgrave, Victoria, Australia). Spectra of samples were recorded from 200 to 2500 nm with a scan rate of 300 nm/min and a step of 0.5 nm. The UV-Vis spectral bandwidth was 2 nm. Estimation of the bandgap energy of the samples was achieved by the derivative peak fitting of the diffuse reflectance UV-visible spectra (DPR) method [35]. Because the derivative decreases in the signal to noise ratio, first, all diffuse reflectance spectra were smoothed using the Savitzky–Golay algorithm with a window of 35 points and a third-degree polynomial. These parameters were chosen because they did not generate a high autocorrelation. Values of the Durbin–Watson statistic were between 2.08 and 2.35. Three Gaussian functions were used for the deconvolution of bands in the range from 300 to 450 nm of the derivative of the diffuse reflectance spectra of the samples. A baseline correction of the Shirley type was applied to the derivative of the diffuse reflectance spectra of four of the five prepared samples before the deconvolution: BT(3.88), BT(4.53), BT(5.59), and BT(6.64). Color parameters L*, a*, and b* of samples were calculated following the CIE 1976 L*a*b* colorimetric method recommended by the CIE (Commission Internationale de l’Eclairage). In this color system, L* is the color lightness (L* = 0 for black and 100 for white), a* represents the green (−)/red (+) axis, and b* is the blue (−)/yellow (+) axis. In addition, chromaticity coordinates in CIE 1931 and CIE L*C*h* systems of all samples were calculated [36]. Additionally, in order to analyze the spectra in the near-infrared region (1800–2175 nm), diffuse reflectance spectra were transformed into a Kubelka–Munk function, and their baselines were manually corrected and then normalized to their maximum.

Diffuse reflectance infrared Fourier transform spectroscopy (DRIFT) was performed in an infrared spectrophotometer from Bruker (model Equinox 55) coupled to an internal diffuse reflection accessory from Praying Mantis. Samples were diluted in KBr at ~5% weight. Spectra were recorded over the range from 4000 to 400 cm^−1^ with a spectral resolution of 4 cm^−1^. Each spectrum was the average of 128 measurements.

Nitrogen adsorption was measured at −196 °C using a surface area analyzer equipment from Quantachrome (model Autosorb-1, Boynton Beach, FL, USA). Before analysis, the samples were outgassed at 80 °C for 24 h, then at 300 °C for 1 h. The specific surface area was determined by the BET method.

Before the measurement of XPS spectra, samples were dispersed in water, irradiated with a tungsten halogen lamp for 30 min, dried with the intention of stabilizing their surfaces [37], and, finally, eroded for 15 s. X-ray photoelectron spectroscopy (XPS) was carried out using a Thermo Scientific instrument (model K-Alpha, Waltham, MA, USA) with a monochromatic Al Kα (1486.6 eV) X-ray source at 12 kV and 40 watts. The X-ray spot size was set to 400 μm and the incident angle was set to 30° to the analyzer. The analyzing chamber was operated with a pressure of 2 × 10^−9^ mbar. Spectra were acquired with a pass energy of 100 eV and a step size of 1.0 eV for the survey spectra. High-resolution spectra were measured with a step size of 0.1 eV and pass energy of 50 eV. The surface atomic compositions (in %) were determined by considering the integrated peak areas of O 1s, Ti 2p, and the respective sensitivity factors.

The photocatalytic performance of the partially reduced titanias was studied for the photocatalytic degradation of the dye methylene blue. The photocatalytic reaction was carried out in a cylindrical glass reactor equipped with a water jacket in which 50 mL of a solution of methylene blue (~10.5 mg/L) was added. The solution was maintained at 293 K with magnetic stirring. Then, 50 mg of the photocatalyst was added to the solution. The system was kept in the dark for 2 h to allow for adsorption equilibrium. Then, the solution was irradiated on the top using an optical fiber connected to a tungsten halogen lamp (150 W). The spectrum of the lamp with the fiber was measured too. Two aliquots of 0.5 mL were collected, one of them of the initial dissolution before the addition of the photocatalyst, and the other 2 h after the adsorption equilibrium. The same aliquot volumes were taken after 0.5, 1, 2, 3, 4, 5, and 6 h, once the irradiation had started.

## 3. Results and Discussion

Compared to the quantities of NaBH_4_ used in the reduction of TiO_2_ reported in the literature [17,38], we used quantities that were considerably less. We studied changes in different characteristics of the samples depending on the amount of NaBH_4_ used for their reduction.

### 3.1. Changes in Structural Characteristics of Reduced Titania Samples

#### 3.1.1. Phases, Lattice Constants, Crystallite Size, and Amorphous Layer

In Figure 1a,b, representative TEM images with different magnifications of the P25 sample are shown. Different morphologies can be observed, such as some distorted circles, squares, and rectangles; also, the presence of several nanoparticles with oval shapes was exposed. These shapes are in accordance with those reported in the literature [39]. Two crystalline phases, i.e., anatase and rutile, were confirmed, respectively, by the inter-planar distances of their planes (101) and (110), as can be seen in Figure 1c,d. In Figure 1e,f, nanoparticles of the BT(4.53) and BT(6.64) samples, respectively, are shown. A layer of amorphous material was regularly observed, especially in the sample treated with the highest amount of reducing agent. The layer thickness was around ~4 to ~8 nm for the BT(6.64) sample. A method other than SEM would have been necessary to observe changes in the morphology of the outer layer formed after the reduction process (see Appendix A).

The composition of phases and the structural and microstructural information of the samples were obtained by the analysis of diffraction patterns. In Table 2, the anatase fraction of the samples are shown, and the rest corresponds to the rutile. For P25, the anatase/rutile fractions values were similar to those reported in the literature. Additionally, other authors have reported 70–85% for anatase, 14–30% for rutile, and a minor quantity (0–18%) of amorphous TiO_2_ [40,41]. In our case, amorphous material was not evident in P25, and it was not considered in the calculated values for the samples shown in Table 2. In general, the quantities obtained by Spurr and Myers’ equation were slightly higher (~1.2%) in comparison to those obtained by the Rietveld refinement, but the trend was similar. The anatase content in the samples BT(3.46) and BT(3.88), prepared with low amounts of reducing agent, did not show any variation (values were similar to P25), but for BT(4.53), BT(5.59), and BT(6.64), a small increment in the anatase fraction was observed.

As mentioned before, the presence of amorphous material was not evident in P25, or BT(3.46) and BT(3.88), but for the other three samples, two small broad unidentified peaks were shown, as can be seen in Appendix A. A gross estimation of the amount of amorphous material in these samples showed that the amorphous phase increased with the increment of the reducing agent: ~0.1% in BT(4.53), ~0.3% in BT(5.59), and more evidently, ~1.2% in BT(6.64) (inset in Appendix A).

Appendix A show a decreasing intensity and peak widening for both phases in samples with the increment of NaBH_4_ used for the reduction. This widening can be attributed to lattice defects created by the chemical reduction process. Common defects are oxygen vacancies and/or Ti^3+^ species; also, a layer of disordered or amorphous material can form on the surface of crystallites [17], with the latter especially having been observed in reduced samples in which a higher quantity of reducing agent was used. Changes in the diffraction patterns and the small increment in the anatase fraction could indicate that rutile has a higher degree of reducibility than anatase. It is known that the length of the Ti-O bonds in rutile is larger than that in anatase, which signifies weaker Ti-O bonds favoring their breakage [42]. Considering the distorted octahedral structure unit (TiO_6_) in rutile and anatase, there are four short and two long bonding distances. In our case, for P25, the short distances for anatase and rutile were 1.938 Å and 1.944 Å, respectively, while the long distances for anatase were 1.982 Å and for rutile 1.997 Å. Both bond types, short and long, are larger in rutile, which may favor their reduction. Lattice parameters and atomic coordinates for anatase and rutile in the samples are shown in Appendix A.

Figure 2a shows a contraction of the cell parameters of both phases compared with those in P25 with the increment of the reducing agent used in the preparation of the samples. One exception was the sample BT(6.64), which used the highest quantity of NaBH_4_. The most significant contraction was for the c parameter in both phases, in comparison to the cell parameter a. As a result of this, cell volume shrinking was observed for almost all samples, but in the case of BT(6.64), an expansion of the cell volume was seen (see Figure 2b). A contraction of 0.26% and 0.10% in the a and c cell parameters, respectively, and a shrinking of the cell volume of 0.66% for a sample of anatase phase were reported in the study of Xia and Chen. Their black titania was prepared from TiO_2_ nanocrystals (white TiO_2_) at a high pressure of 20 bar in H_2_ atmosphere at 200 °C for 5 days [31]. Different results were obtained by Naldoni et al. [13], where one experiment showed for the anatase phase a diminution of 0.03% in the a parameter and an expansion of 0.14% in the c parameter with respect to white TiO_2_. The cell volume exhibited an expansion of ~0.06%. Those samples were prepared from commercial amorphous powder of TiO_2_ calcined at 200 °C. White TiO_2_ was obtained after calcination at 500 °C and black TiO_2_ by a reduction in H_2_ flow, and in both cases, the thermal treatments lasted 1 h. In the literature, it has been reported that diffraction peaks shift to higher [43] or lower [18] diffraction angles, thereby making a contraction or expansion of the lattice possible.

Variations in the crystallite size and microstrain with the amounts of reducing agent are shown in Figure 2c,d, respectively. While the determination model for a wide crystallite size distribution is limited [44], it gives us an idea about the size distribution, which is in agreement with the TEM images. In TiO_2_ P25, the anatase crystallites (~23 nm) were smaller than those of rutile (~36 nm). Similar crystallite sizes have been reported for TiO_2_ P25, varying for anatase from 20 to 28 nm and for rutile from 31 to 47 nm [35,45,46]. Commonly, the size of the anatase crystallites is smaller than 30 nm [40]. In the present study, almost all reduced samples presented the same anatase crystallite size, which was similar to the anatase in TiO_2_ P25. Only the BT(6.64) sample revealed a small diminution of ~3 nm in the crystallite size. Rutile changes were more significant: diminutions in the crystallite size for the samples BT(5.59) and BT(6.64) were around ~9 and ~18 nm, respectively. These results were in agreement with the layer thickness of amorphous material observed in TEM and described above. Additionally, the size in two directions is shown in Appendix A. For TiO_2_ P25, anatase crystallites were slightly anisotropic, but rutile crystallites were anisotropic. The anatase phase in TiO_2_ P25 showed microstrain, and this could have been due to defects created during the synthesis process of TiO_2_ P25. Unexpectedly, in most samples of our blue titania, microstrain for anatase decreased with the increment of the reducing agent. For rutile, only the BT(6.64) sample showed microstrain (see Figure 2d).

Rutile exhibits remarkable structural changes, suggesting that rutile is preferentially reduced over anatase, at least under our experimental conditions. In general, the prepared samples with a $w_{NaBH_{4}}$ < 5% suffered slight changes, whereas samples with a $w_{NaBH_{4}}$ > 5% clearly showed changes in crystallite sizes. This suggests that the changes prompted by the use of small quantities of reducing agent preferably occurred on the surface, but with the increment of $w_{NaBH_{4}}$, changes occurred additionally in the volume of the crystallite. The cases where the crystallite size decreased could indicate the formation of an amorphous or disordered layer of reduced TiO_2_ around a crystallite core, forming a particle-type crystalline-disordered core-shell structure, as was observed in TEM. The results are in agreement with the presence of amorphous material in our diffraction patterns for samples prepared with higher quantities of reducing agents. The formation of amorphous or disordered material layer around the crystalline core has also been reported for materials prepared by the mixing and grinding of TiO_2_ P25 and NaBH_4_ [17,18,38].

Mechanisms by which defects, such as oxygen vacancies, Ti^3+^ species, layers of disordered or amorphous material, etc., emerge, are unclear and depend on the type of synthesis/preparation. Further, not all these defects are always present in black TiO_2_. It has been reported that probably during the reduction process, oxygen of the TiO_2_ lattice can be removed via liberation of water for a hydrogen reductant, or oxygen molecules for a non-hydrogen reductant, forming a surface oxygen vacancy [7]. In addition to this, it has been reported that in anatase, and similarly in rutile, when an oxygen vacancy is created, the titanium atoms shift slightly away from the vacancy in order to strengthen their bond with the rest of the lattice. On the other hand, the neighboring oxygen atoms shift slightly inwards toward the vacancy to fill the empty space [24]. This oxygen vacancy can move to the subsurface or even into the crystallite volume if the conditions allow it. If this happens, a lattice reconstruction occurs in order to relax the non-equilibrium lattice strain [7]. If the oxygen vacancies concentration is high enough, it can cause lattice disorder. Alternatively, if the lost oxygen leaves a pair of electrons in the TiO_2_ matrix, these electrons can be trapped and form oxygen vacancies with different charges and/or react with Ti^4+^ to form Ti^3+^ [7]. Then, the relaxation of the lattice, i.e., changes in the cell volume and changes in the crystallite size, could explain the microstrain decrease in the anatase phase. For rutile TiO_2_, it has been reported that lattice distortion depends on the type of defect, such as oxygen vacancies and Ti interstitials, and their relative concentration as well. It has been proposed that the interstitial Ti^4+^, F^+^ (oxygen vacancy plus one electron), and neutral oxygen vacancy defects are primarily responsible for lattice expansion, whereas the electrostatic attraction between interstitial Ti^4+^ and interstitial O^2−^ defects causes the lattice contraction in the undoped TiO_2_ nanostructures [47]. In our case, the probable increment of the Ti^3+^ species concentration after the reduction process in the BT(6.64) sample could have favored cell expansion, because the ionic radius is larger in Ti^3+^ than in the Ti^4+^ ion.

According to the Wulff construction, crystals are dominated by thermodynamically stable facets. However, they break down under various conditions [31]. Furthermore, in the present study, crystallite or crystalline domain shapes of both phases were studied. For this, half of the mean column length sizes along different directions obtained by Rietveld refinement were considered. The first approach was the arbitrary selection of surface planes with low surface energy for the construction of the crystal shapes. Planes {101}, {100}, and {001}, which are commonly found in anatase [31], were selected; for rutile, the considered surface planes were {110}, {100}, and {101} [48]. The crystallite shapes of anatase and rutile are shown in Appendix A for two samples and the area of the surface planes in Appendix A. The exposed area of anatase of the {001} surface in P25 was around 22.8%, a value similar to the reported one of ~20–22.73% in other studies [49,50]. Surface plane areas for anatase in the P25 and BT(6.64) samples were not different. However, for the rutile phase, there were changes: a diminution in the area of the {101} and an increase in the area of the {100} surface planes occurred. This approach for the construction of the crystalline domain was expanded by considering all sizes obtained by Rietveld refinement, along with different directions. These shapes are shown in Appendix A and the surface plane areas in Appendix A. An important finding was that changes in the surface plane areas were relevant for the rutile phase. The BT(6.64) sample showed a different crystalline facet than in TiO_2_ P25, which could be thought to be the crystalline core of a rutile phase particle.

#### 3.1.2. Surface and Bulk Defects

Additional structural information was obtained by Raman spectroscopy. Representative spectra of all samples are shown in Figure 3a. In general, the anatase phase spectrum was more intense than the rutile spectrum in all measurements, because the anatase phase is predominant. For the anatase phase, there are six Raman active modes: 3 E_g_, 2 B_1g_, and 1 A_1g_ [13]. The most intense band in the spectrum corresponding to the E_g_ mode at around ~144 cm^−1^ was examined. An outstanding outcome was that blue shifting and widening of the band were observed (as shown in Figure 3a,b) with the reducing agent’s increment; both effects can be caused by defects (nonstoichiometry, etc.) and a decrease in crystalline domain size [13,17]. According to Zhu, defects led to Raman frequency and bandwidth changes, because of the shortening of the correlation length of the phonons [51].

Two slopes are visible in Figure 3b for changes in the E_g_ mode’s frequency position and width, one for $w_{NaBH_{4}}$ ≤ 4.53% and the other for $w_{NaBH_{4}}$ ≥ 4.53%. This could have been due to different types of defects being generated in crystallites after the reduction process. Prepared samples with small quantities of reducing agent preferably presented surface defects, but with the increment of $w_{NaBH_{4}}$, defects occurred additionally in the volume of the crystallites. In addition, the increment of $w_{NaBH_{4}}$ could have induced a possible saturation in the line widening and shifting the position of the band. The Raman frequency shifted from 143 to 148 cm^−1^, and the crystalline size domain went from ~23 to 20 nm. There was a blue shifting of ~5 cm^−1^ for the BT(6.64) sample compared with TiO_2_ P25. Shifting in the Raman frequency caused by phonon confinement reported for crystallite sizes that went from 25 to 15 nm, 28 to 19 nm, and 20–16 nm was only ~1, ~1, and ~1.5 cm^−1^, respectively [51,52,53]. The shifting of the Raman frequency was larger in our case. The same occurred with the bandwidth—it was broader in our result—so this may not have just been a size effect. Naldoni et al. reported that the occurrence of the finite size effect was discarded, or its contribution was minor, and the large blue shifting and the band broadening were caused by a structural disorder [13] and/or additional defects. A linear correlation between the Raman frequency and width was found (see Figure 3c), and it was very similar to the relation caused by phonon confinement that has been reported in nanocrystalline materials [54]. The abovementioned suggests that the slope of the linear correlation (excluding P25), the presence of two slopes in the E_g_ mode’s frequency position, and its width for the anatase phase as a function of the amount of NaBH_4_ (see Figure 3b,c, respectively) were due to the type of defect and its distribution, and these were different from those presented only by the size effect of the crystalline domain.

### 3.2. Optical Characteristics

#### Reflectance, Absorption, Bandgap Energy, and Color

Optical characteristics were studied by diffuse reflectance spectroscopy in the ultraviolet/visible/near-infrared regions (see Figure 4). Reflectance spectra of the samples are shown in Figure 4a. For P25, the absorption edge was around 400 nm. The UV region was where P25 had minor reflectance, and the high values were in the visible and near-infrared regions. With the increment of the reducing agent, the prepared samples showed a decrease in their diffuse reflectance in the visible and, especially, near-infrared regions. Similar behavior has been observed in samples of TiO_2_ reduced with high concentrations of NaBH_4_ [17,38] or CaH_2_ [55].

Diffuse reflectance spectra were transformed into the Kubelka–Munk function to examine the contribution of the reduction process to samples’ absorption spectra (see Appendix A), and the P25 spectrum was subtracted from all reduced samples (see Figure 4b). Then, it was found that there was a contribution at 3.45 eV (359 nm). Interestingly, the intensity of the contribution at 3.45 eV was very similar for all reduced samples. This band could have even been due to isopropyl alcohol residues on TiO_2_ remaining after the impregnation and the subsequent thermal treatment. This statement is based on the fact that the band was observed in the sample prepared without the addition of NaBH_4_ (not shown), but an effect caused by NaBH_4_ cannot be discarded.

With respect to visible and near-infrared regions, an increase in absorption was observed where a new maximum was found around 0.95 eV (1305 nm or 7662.8 cm^−1^) and a wide shoulder between 1.5 to 2.5 eV. Mishra et al. reported that vacancies of oxygen or titanium in TiO_2_ induce the appearance of low energy peaks in the optical absorption spectrum. Oxygen vacancies originate new bands between 1.5 and 2.5 eV. Titanium vacancies can also originate bands between 0 and 2.0 eV with a higher intensity in the energy range < 1.0 eV [56]. However, oxygen vacancies are energetically more favorable and thus more likely to be created. In the case of TiO_2_, oxygen vacancy leads to a donor level below the conduction band. In addition, oxygen vacancy also gives rise to incomplete/broken bonds; therefore, weakly bounded electrons change the oxidation state of Ti^+4^ toward Ti^+3^ [57]. Other authors have revealed that a favorable process during the reduction of anatase is the formation of interstitial titanium [58]. Therefore, the new features in the absorption spectra of the reduced samples were due to defects, most probably oxygen vacancies, but interstitial titanium should be considered too, especially for the BT(6.64) sample. Zhu et al. pointed out that the amorphous shells of hydrogenated titania do not necessarily improve light absorption [10]. In that study, the authors suggest that hydrogen atoms doped into the crystal lattice of TiO_2_ were responsible for the increased visible and near-infrared light absorption. Therefore, an amorphous shell is not essential for the formation of colored titania, but different defects in TiO_2_ are important to improve the absorption of light in the visible and NIR regions. In this sense, further investigations are needed to identify with precision the contribution of different defects in reduced TiO_2_ to its absorption spectrum.

An estimation of the bandgap energy was carried out. There are different methods to estimate the bandgap energy of semiconductors, but sometimes these are difficult to apply or not appropriate to be applied, particularly when there is a semiconductor mixture [35] and/or when there exists more than one optical absorbing entity [59]. In our case, we used the derivative peak fitting of the diffuse reflectance UV-visible spectra (DPR) [35], because the samples were a blend of semiconductor polymorphs and the reduced samples had a band tail in the low-energy region. With the aid of the deconvolution of bands, this is a useful method, and applicable to TiO_2_, in which the bandgap energy was found in the local maxima of the first derivative of diffuse reflectance spectra. The estimated values of the bandgap energy obtained by DPR were related to those estimated by other different methods, for example, by the Tauc plot [60]. We observed that this method reduced the influence of the large band tail of the spectra in the reduced samples for the determination of the bandgap energy.

DPR analysis allowed for the identification of three local maxima in all samples. The value trends of the bandgap energy for anatase and rutile are shown in Figure 4c for the set of samples. The derivatives of the diffuse reflectance spectra are shown in Appendix A, where three local maxima found in P25 are indicated. The values of the bandgap energy found for the rutile and anatase phases were 3.14 and 3.39 eV, respectively. There was a third maximum localized at 3.61 eV, but it was not assigned. Bandgap energy values were similar to those reported for TiO_2_ P25: 3.17, 3.43, and 3.70 eV for rutile, anatase, and the unassigned maximum, respectively [61]. Moreover, 3.13 and 3.33 eV for rutile and anatase, respectively, have been reported for TiO_2_ P25 without the consideration of a third maximum [35]. A hypothesis suggested by Apopei et al. about the maximum of high energy (3.61 eV) is that this could be due to the presence of amorphous TiO_2_ [61]. Unfortunately, our analysis was not conclusive about the relationship between the band at the high energy value of 3.61 eV and the presence of amorphous TiO_2_. First, and importantly, the intensity of this band decreased in the reduced samples in comparison to TiO_2_ P25. However, among the reduced samples, the relative intensity of this maximum increased with the increment of NaBH_4_ (see Appendix A). In addition, the presence of amorphous TiO_2_ in P25 was not evidenced, but its occurrence and increase with the increment of the NaBH_4_ amount was detected by XRD. Therefore, more studies are necessary to understand the origin of the high-energy maximum. For reduced titania samples, values of the bandgap energy obtained by DPR method have not been reported in literature.

With regard to changes in the bandgap energy, it can be seen that the value for the unassigned maximum increased by about ~0.8 eV, and then it decreased and remained approximately constant. For anatase, it first decreased and then increased, and likewise for rutile. Absorption in the UV region shorter than 400 nm can be attributed to the intrinsic bandgap absorption of TiO_2_. The reduction process changed the inflection point in the diffuse reflectance spectra in the UV region, shifting it to a lower wavelength; thus, the bandgap energy estimated by the DPR method increased. When reduction occurred, the surface of the TiO_2_ particles was reduced. Therefore, different defects were created, and with the increment of the reducing agent, a layer of disordered or amorphous material was formed. Then, the crystalline core size decreased while the bandgap energy increased, particularly for samples in which a high quantity of reducing agents was used. However, reduced samples were absorbed in the visible region. This apparent contradiction can be clarified by considering that the large band tail absorption originated by different defects in the reduced samples. Similar considerations have been proposed by Teng et al. for black TiO_2_ prepared by hydrogen plasma-assisted chemical vapor deposition, where oxygen vacancies were principally considered for a scheme of the density of states, potentially explaining the absorption spectrum [62]. In addition to this, in our case, titanium vacancies could have been involved, because spectra suggested their presence: for instance, the absorption band around 0.95 V in samples prepared with $w_{NaBH_{4}}$ ≥ 3.88%. During the reduction process, titanium vacancies could have formed as a consequence of interstitial titanium formation, which is when titanium ions leave their original positions in the lattice to occupy interstitial positions. As mentioned above, when an oxygen vacancy is created, the titanium atoms shift slightly away from the vacancy in order to strengthen their bond with the rest of the lattice. In general, effects in the absorption spectra are caused by defects associated with oxygen and titanium atoms, and the latter especially occurred in the reduced samples in which a larger quantity of reducing agent was used. As a consequence of the mentioned defects, new interband states would have been created on the state’s density near the valence and conduction band.

The color of the samples was studied too. P25 turned from white to dark blue with the increment of the amount of the reducing agent. Figure 4d shows the colorimetric parameter values using the CIE L*a*b* coordinates. It is noticeable that lightness, L*, decreased from 100 to ~48% with the increment of the reducing agent. In general, for the reduced samples, the a* and b* parameters shifted toward negative values. These results indicate that the reduced samples showed green and blue components in their color. The b* parameter showed a more significant change than a*, so the samples showed a blue color, inspiring the nomination of the material in the title of our investigation. Additional colorimetric information in the CIE 1931 and CIE L*C*h* systems is shown in Appendix A.

### 3.3. Surface Characteristics

#### 3.3.1. Loss of Hydrophilicity

A close look at the near-infrared region provides additional information. Appendix A shows different sections of the near-infrared region of the spectra. Negative bands can be observed at 1360 nm (7353 cm^−1^) and 1384 nm (7225 cm^−1^) in Appendix A and at 2213 nm (4519 cm^−1^) and 2256 nm (4433 cm^−1^) in Appendix A. The last two bands are assigned to the combination (ν + δ) of the stretching (ν) and bending (δ) modes of the non-bonded and bonded silanol groups [63,64]. Both bands were due to the quartz window of the sample holder. The band at 1384 nm (7225 cm^−1^) was due to the overtone (2ν) of the stretching mode (ν) of the silanol group.

The samples P25 and BT(3.46) showed a band at 1456 nm (6868 cm^−1^) and another weak broad band at 1768 nm (5656 cm^−1^) (see Appendix A). The first band was assigned to the combination band of the symmetric and asymmetric stretching (ν_sym_ + ν_asym_) vibration modes of water. The second band can be attributed to a combination (ν_sym_ + ν_asym_ + ν_L_) of stretching symmetric, asymmetric, and ν_L_ vibration modes of the water molecule [65] (ν_L_ was not defined in the reference, but it has been postulated to be an “intermolecular mode”). Additionally, the increment of the absorption towards 2500 nm could have been due to the presence of the water [66] (see Appendix A). These bands disappeared in the reduced samples in which a $w_{NaBH_{4}}$ > 3.46% was used for their preparation.

The spectral region of 1800–2175 nm (see Appendix A) offers information about the partially reduced TiO_2_ surface interaction with physisorbed water molecules. It provides information about the state of molecularly adsorbed water on TiO_2_ [67]. For the P25 and BT(3.46) samples, very wide absorption bands were observed at 1885 nm (5305 cm^−1^) and 1937 nm (5163 cm^−1^) (see Appendix A). These bands were assigned to the combination (δ + ν_asym_) of the bending and asymmetric fundamental vibration modes of physisorbed water molecules [63]. For the reduced samples ($w_{NaBH_{4}}$ > 3.46%), these bands suffered an important variation. The (δ + ν_asym_) mode was intense with respect to the other combination modes and did not suffer from significant overlap with any components due to surface hydroxyl groups. Furthermore, the contribution of the H_2_O ν_asym_ mode makes this signal sensitive to the interactions experienced by water molecules [68]. The water molecule works as a hydrogen bond acceptor or donor. The donor hydrogen bonds can be referred to as “active H-bonds” and the acceptor hydrogen bonds can be referred to as “passive H-bonds”. One water molecule can have up to four hydrogen bonds (two active and two passive hydrogen bonds). Because the active hydrogen bond is more significant than the passive, the chemical state of water can be reflected by the number of active hydrogen bonds [63]. However, researchers have pointed out that it is difficult to distinguish the two different hydrogen bonds by spectroscopic measurements because the hydrogen bond networks can be easily reconstructed in a short time even at room temperature [69].

Commonly, bands at 1885 and 1937 nm are deconvoluted into four components [63,69,70]. These components are S_0_, S_1_, S_2_, and S_n_, with these having wavelengths of 1884, 1928, 1989, and 2066 nm (5038, 5187, 5028, and 4840 cm^−1^), respectively, which were observed in TiO_2_ P25 [63,70]. The components are assigned to different structures of the physisorbed water molecules which depend on the number of intermolecular hydrogen bonds. The component “S_0_” is assigned to adsorbed water molecules acting only as hydrogen-bond acceptors, forming the outermost shell of the H_2_O surface multilayer that is called “free water”. The component “S_1_” is ascribed to water molecules with one hydroxyl group involved in H-bond donation, and with or without bonding to the oxygen. Component “S_2_” is attributed to water molecules involved in H-bond donation on each hydroxyl group, with or without bonding to the oxygen. Moreover, the component “S_n_” is assigned to water molecules with two active and one (or two) passive H-bond(s) (polymeric chained H_2_O molecules) [63,69]. All these polymeric chained H_2_O molecules (S_1_, S_2_, and S_n_) are called “hydrogen-bonded water”.

After the conversion of the diffuse reflectance spectra into the Kubelka–Munk function, the signals of the reduced samples ($w_{NaBH_{4}}$ > 3.46%) in the region of 1800–2150 nm were clarified (see Appendix A). The baseline-corrected and normalized spectra of the prepared samples in the near-infrared region are shown in Figure 5, whereby the positions of the components S_0_, S_1_, S_2_, and S_n_ [63,69,70] are indicated with dashed lines. The reduced samples ($w_{NaBH_{4}}$ > 3.46%) exhibited changes in the shape and the intensity of the bands related to physisorbed water.

Interestingly, a new band at 1853 nm (5397 cm^−1^) appeared in the reduced samples ($w_{NaBH_{4}}$ > 3.46%). Unfortunately, this latter absorption feature could not be assigned with some certainty. We are carrying out additional studies to improve our understanding of this band.

On the other hand, it is evident that the affinity of the surface upon water molecules lowered with the increment of the reducing agent used in the preparation of the blue titania samples. There was a decrease in the relative intensity and full width at half maximum (FWHM) of the bands in the reduced samples when they were compared with those of P25. This suggests that the amount of physisorbed water decreased in comparison with the unreduced TiO_2_ P25. In addition, absorption in the region of the S_2_ and S_n_ bands decreased and transformed into a flatted long tail extended up to 2150 nm in the reduced samples, which implies that the outer surface areas of the water clusters on the surface also decreased.

#### 3.3.2. Decrease in Hydroxyl Groups

In general, the reduced samples showed the lowest diffuse reflectance values in the infrared region (not shown). Displaced infrared spectra are depicted in Figure 6. Spectra exhibited a band intensity decrease in the 400–800 cm^−1^ region, which was associated with the Ti-O-Ti stretching vibration mode in the crystal of TiO_2_ (see Figure 6a). This decrease can be attributed to different defects created by the reducing process (like oxygen vacancies, Ti^3+^, etc.), whose concentration increased with the amount of reducing agent used. However, these bands did not disappear entirely even in BT(6.64), thus demonstrating the preservation of the TiO_2_ crystalline core.

As seen in Figure 6a, the sharp band at 1635 cm^−1^ was assigned to bending modes of the physisorbed water molecule. The broad band at 2700–3700 cm^−1^ was due to stretching modes of the surface hydroxyl groups of TiO_2_ and the presence of water molecules. With the increment of the reducing agent, the progressive disappearance of these bands can be observed. Thus, the reducing process decreased the number of surface hydroxyl groups on TiO_2_, creating some kind of surface hydrophobicity; under this circumstance, the interaction of the surface with water molecules is less favorable. These observations show good correspondence with the results obtained from near-infrared measurements. The results show some similarities with the effects caused in TiO_2_ powder when it was photoreduced in the absence of O_2_. TiO_2_ powder turned from white to blue-gray when it was photoreduced. It should be mentioned that the photoformed electrons were trapped on the Ti^4+^ to produce the Ti^3+^ sites, while the majority remained in the conduction band. The photoreduced TiO_2_ surfaces can be represented as “electron-rich surfaces” or “negatively charged surfaces”. Previous researchers have also mentioned that photoreduced TiO_2_ surfaces are very stable in the absence of O_2_ for long periods and hardly interact with H_2_O molecules [64]. The same types of defects—oxygen vacancies and/or Ti^3+^ species—could have formed on the surface of TiO_2_ by the reduction process with NaBH_4_. It is known that NaBH_4_ can act as a reductant directly or hydrolyze to release the reductive H_2_ [71]. The results suggest that during the reduction process in the nitrogen atmosphere, oxygen atoms of hydroxyl groups and TiO_2_, both from the surface and the lattice, were consumed, whereas there was a minor possibility this occurred from chemisorbed and/or physisorbed water. The existence of Ti^3+^ in the TiO_2_ lattice matrix can be expected to generate oxygen vacancies to maintain the electrostatic balance according to the following reported chemical equation [72]:4Ti^4+^ + O^2−^ → 4Ti^4+^ + 2e^−^/□ + 0.5O_2_ → 2Ti^4+^ + Ti^3+^–□–Ti^3+^ + 0.5O_2_
where □ is a vacancy originating from the removal of O^2−^ in the lattice. Therefore, the reduced TiO_2_ samples may have had negatively charged surfaces, which explains the low affinity of these surfaces toward water molecules. Strongly polarized water molecules (due to the electronegativity of the oxygen atoms) hardly interact with such negatively charged surfaces and similarly with photoreduced TiO_2_ surfaces.

Bands at 2895 and 2986 cm^−1^ were assigned to C-H stretching vibrations of methylene and methyl groups [73] due to hydrocarbon moieties from the atmosphere (see Figure 6b).

In Figure 6b, the band at 1047 cm^−1^ is ascribed to the presence of the carbonate group [74] generated from adsorbed CO_2_ molecules. Additionally, the bands at 1420 cm^−1^ and 1227 cm^−1^ can be assigned to carbonate groups [74,75].

Bands at 1227 and 1535 cm^−1^ could not be attributed with certainty, and these were present in all samples, including TiO_2_ P25 (see Figure 6b). The band at 1530 cm^−1^ was assigned to the formation of surface disorders of hydrogenated TiO_2_, which was reduced in a continuous flow of pure hydrogen under a thermal treatment at 500 °C for different periods. This band increased with the increment of the duration of the thermal treatment in the hydrogen flow [76], but it was also present in pristine TiO_2_.

Sharp noises at around 3600–4000 cm^−1^, as seen in Figure 6b, were due to water vapor [64], but the intensity of some signals in this region increased with the amount of reducing agent used. Several sharp bands can be observed at 3847, 3754, 3682, 3630, 3587, 3505, and 3415 cm^−1^. The band at 3682 cm^−1^ was assigned to tetrahedrally coordinated vacancies and is designed as _4_Ti^4+^–OH (tetrahedrally coordinated) in TiO_2_ [77]. Bands at 3847 and 3754 cm^−1^ can be attributed to _6_Ti^3+^–OH (octahedral vacancies) [37,76,77,78], which were visible in our reduced samples.

#### 3.3.3. Diminution of OI/Ti Ratio

The set of samples was studied by XPS. Before the measurement of XPS spectra, the surface of the samples was stabilized. This consideration was made because photoreduced TiO_2_ surfaces irradiated with UV light in the presence of O_2_ are immediately oxidized, and the surface-trapped electrons are scavenged by O_2_ [64]. We hoped that our samples would show a surface that was very similar to the one during the photocatalytic test but were expectant of the fact that the samples were eroded by 15 s. Characteristic O 1s and Ti 2p peaks of all samples are shown in Figure 7a,b, respectively. The peaks of the reduced samples were wider than in TiO_2_ P25. This suggests the presence of oxygen vacancies and Ti^3+^ species, which is in agreement with the infrared spectra above. Survey spectra and deconvolution of O 1s spectra are shown in Appendix A. In Figure 7c, the OI/Ti ratio is presented; the OI-species is called structural oxygen. OI-species were related to oxygen vacancies in the TiO_2_ lattice. The calculated values were not the real stoichiometric ratio of the titanium dioxide because the titanium amounts were overestimated by our not distinguishing between structural and superficial titanium [79]. While the samples were eroded, the OI/Ti ratio showed the trend of the presence of oxygen vacancies in the samples. Figure 7c demonstrates that with the increment of the amount of the reducing agent used, the OI/Ti ratio decreased due to the increment of the oxygen vacancies.

#### 3.3.4. Texture Preservation in Blue Titania Samples

Textural characteristics of the samples were determined by nitrogen adsorption/desorption isotherms. Figure 8 shows the experimental isotherms for the set of prepared samples. The isotherms of the blue titania samples had the same shape and were similar to the shape of TiO_2_ P25. They exhibited a type II adsorption isotherm according to the updated IUPAC recommendations [80], which is typical for non-porous or macroporous solids. The type H3 narrow hysteresis loop indicates the presence of false particles’ macropores, which may reflect interglobular cavities resulting from the agglomeration of nanoparticles.

The overall shape of both the adsorption-desorption isotherms and hysteresis loops of the reduced samples were preserved if they were compared with P25, which indicates that the texture of the reduced samples was not altered.

The values of the BET-specific surface area (S_BET_) and total pore volume are listed in Appendix A. S_BET_ varied within 46 and 59 m^2^ g^−1^, values that are practically alike if we consider the precision of the method. The Barret–Joyner–Halenda pore size distribution curves exhibited a maximum in the mesopore size range (16 nm) for all samples (except for BT(3.46)), indicating the similarity of the nanoparticle agglomerates in the sample series.

Therefore, the changes in the surface properties of the reduced samples studied by near and mid-infrared spectroscopy were principally due to the chemical reduction with NaBH_4_, which did not significantly alter the texture of the TiO_2_ samples.

### 3.4. Low Degradation of Methylene Blue by Blue Titania

Photocatalytic activity was studied using as a degradation model the methylene blue (MB) dye, which is a cationic dye [81]. The kinetic parameters of the photocatalytic degradation of the dye were obtained according to a pseudo-first order reaction. Kinetic curves and values of the constant rate (k_app_) are shown in Figure 9. The inset in Figure 9 shows that the TiO_2_ P25 sample exhibited the highest k_app_ value and that it decreased with the increment of the reducing agent in the preparation of the samples set. This result contrasts with that observed with TiO_2_ P25 reduced with NaBH_4_ by a different methodology, which presented higher photocatalytic activity in the degradation of methyl orange, an anionic dye [17].

We observed the adsorption of MB in dark conditions and the trend was for this to increase with the degree of loss of hydrophilicity in the reduced samples. MB removal percentage by photocatalysis varied between 89% and 76% for BT(3.46) and BT(6.64), respectively. Values of adsorption by all samples and photolysis are indicated in Appendix A. Although at first sight, these results may appear to be of no practical value, they strengthen knowledge about the role of the hydroxyl groups, the amorphous layer, a deficient oxygen surface in the generation of hydroxyl radicals.

Considering the increase in the overlapping of the emission spectrum of the lamp used in the photocatalytic test and the absorption spectra of the prepared samples (Appendix A), it would be expected that the reduced samples exhibit a higher photocatalytic activity. However, that was not the case. As was mentioned, absorption in the UV-Vis region by the reduced samples is due to the presence of defects. In literature, there exist contradictory statements on the role of Ti^3+^ species/oxygen vacancies in photocatalytic activity. Authors have reported that defects act as recombination centers for the photogenerated electron-hole pairs and cause a decrease in photocatalytic efficiency [24]. On the other hand, it is stated that oxygen vacancies improve charge separation [12]. These studies might seem counterintuitive. Therefore, to understand the behavior of the photoinduced charge carriers, their generation and separation in reduced samples, origin in the crystalline core and mobility through the amorphous layer, and the nature of the defects and concentration, among other characteristics, should be considered. It is common not to address the surface reactivity of black TiO_2_ towards water molecules or other substances [12], and this frequently implies the presence of a disordered or amorphous layer. In this study, we showed for the first time spectroscopic evidence of changes in the surface properties of reduced samples and these changes’ possible effects on the photocatalytic properties of these samples. We demonstrated that the loss of surface hydroxyl groups implied a decrease in the surface hydrophilicity in the reduced samples with respect to P25. This modified the reactivity of the surface towards water molecules. In addition, the photocatalytic activity of TiO_2_ was closely related to the surface hydroxyl groups. There is evidence that surface hydroxyl groups play an important role in the generation of hydroxyl radicals [82]. Therefore, the reduced samples of blue titania show less photocatalytic activity compared with the starting material, TiO_2_ P25.

## 4. Conclusions

The characteristics of the TiO_2_ P25 reduced to different extents by the use of five concentrations of the reducing agent NaBH_4_ were studied. Rutile showed a higher degree of reducibility than anatase, at least under our experimental conditions. Blue titania samples with smaller quantities of the reducing agent were principally affected on the surface. The use of higher concentrations affected the particles in the bulk. At a level of 6.64% NABH_4_, the amount of Ti^3+^ created caused an expansion of the cell volume in both phases. Additionally, for these samples, a layer of disordered or amorphous TiO_2_ was created on the surface of the particles, forming a particle-type crystalline-disordered core-shell structure.

Optical properties were studied, and, for the first time, the colorimetric parameters of reduced titania nanoparticles were obtained. The set of samples showed the absorption of light in the UV–Vis–NIR regions of the electromagnetic spectrum. Small quantities of reducing agents produced a decrease in the bandgap energy, while high quantities produced an increment. This suggests there was a decrease in the size of the crystalline core. Both XRD and DRS UV–Vis–NIR studies indicated the presence of interstitial Ti^3+^ in samples prepared with NaBH_4_ > 3.46%.

The results showed that, during the reduction process, oxygen atoms of hydroxyl groups and of the surface and lattice of TiO_2_ were consumed, whereas there was a minor possibility this occurred from chemisorbed or physisorbed water. XPS spectroscopy showed a decrease in the amount of structural oxygen as a consequence of the reduction. Near-infrared measurements showed that the reduction in the surface was associated with a decrease in the hydroxyl groups and that this triggered a decrease in the surface hydrophilicity. In addition, the presence of _6_Ti^3+^–OH (octahedral vacancies) in the reduced samples was suggested by Mid-DRIFT. The reduction process did not affect the textural properties.

The blue titania samples showed a decrease in the photoactivity in the degradation of the dye methylene blue, despite the overlapping of the emission spectrum of the lamp used in the photocatalytic test and the higher absorption observed in comparison with TiO_2_ P25. The results suggest that the reduction process changed the reactivity of the surface with the molecules of water and dye. In dark conditions, there was an increment in the adsorption of the dye, but at the same time, there was a decrease in the affinity of water molecules for the reduced sample surfaces. In addition, the important decrease in the hydroxyl groups on the surface of TiO_2_ could have affected the photocatalytic properties associated with the generation of the hydroxyl radical. Consequently, the reduced samples showed less photocatalytic activity compared with P25. Therefore, the changes in the surface properties due to the reduction process and their possible effect on the photoactivity related to water (polar) and/or dye (MB) ionic molecules have been addressed for the first time.

## Figures and Tables

**Figure 1 nanomaterials-12-01501-f001:**
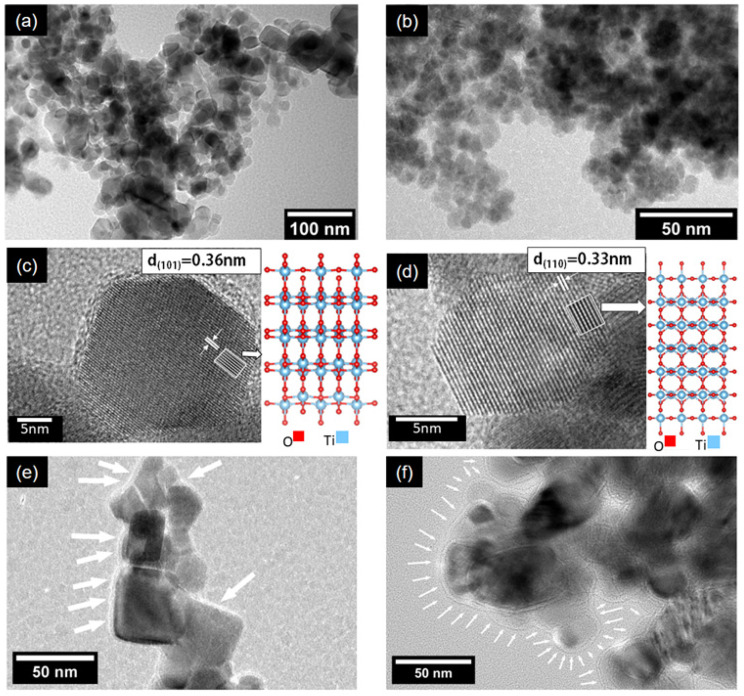
TEM images of the samples. P25 particle morphology at low magnification in (**a**,**b**). Anatase and rutile crystals in (**c**,**d**), respectively. Respective crystalline structure used to simulate the HRTEM images, right side of (**c**,**d**). Distances corresponding to the planes (101) and (110), respectively, indicated in a white box. In (**e**,**f**), particles of the reduced samples, BT(4.53) and BT(6.64), respectively. The layer of amorphous material observed in reduced samples is indicated with white arrows.

**Figure 2 nanomaterials-12-01501-f002:**
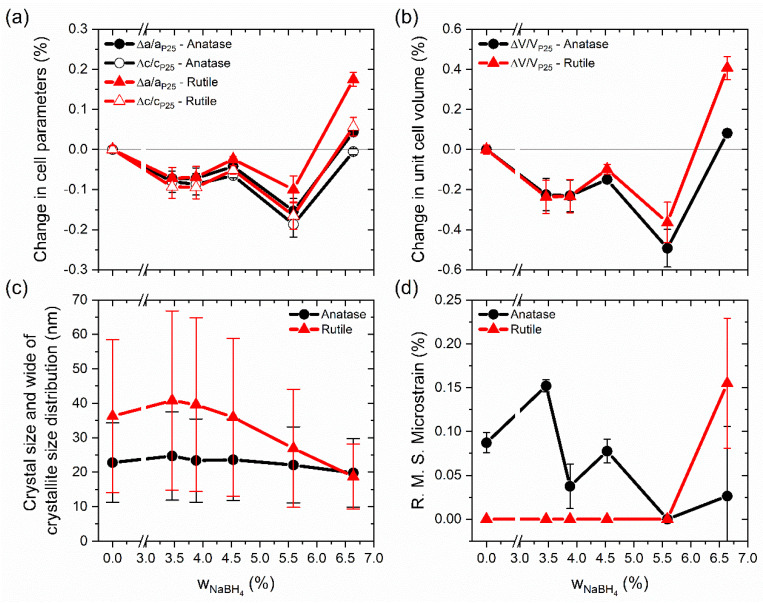
Changes in (**a**) cell parameters and (**b**) unit cell volume with respect to TiO_2_ P25. In (**c**), crystal size corresponds to a volume-weighted mean sphere diameter and distribution width (bars); (**d**) shows microstrain variation.

**Figure 3 nanomaterials-12-01501-f003:**
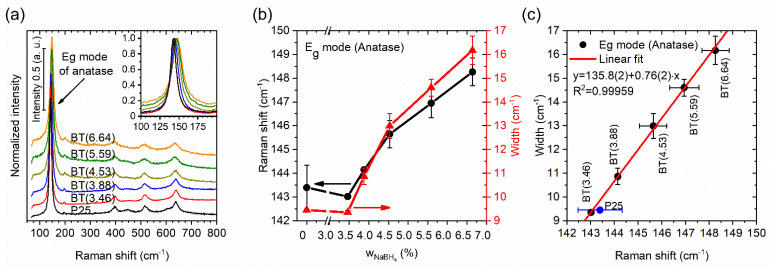
(**a**) Raman spectra of the samples, inset: magnification of the E_g_ mode. (**b**) Raman shift and width of the E_g_ mode as function of the amount of reducing agent used. (**c**) Relationship between Raman width and shift of the E_g_ mode.

**Figure 4 nanomaterials-12-01501-f004:**
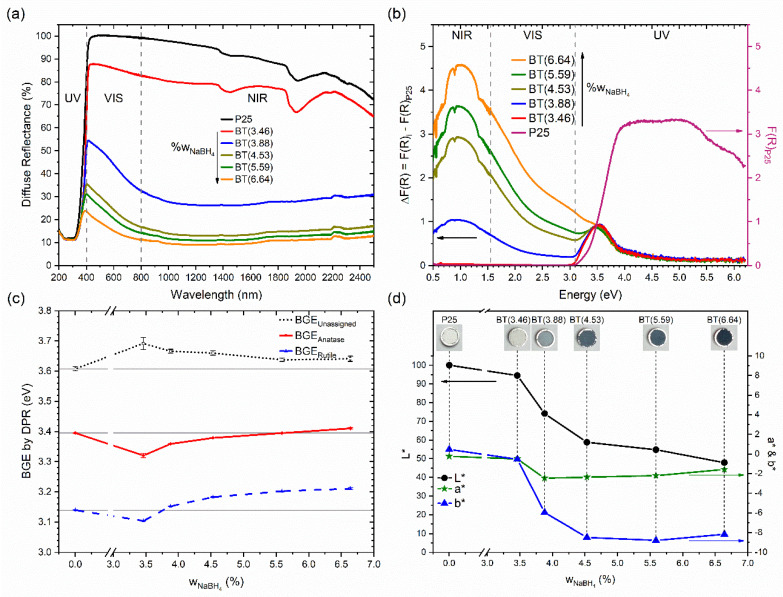
(**a**) Reflectance spectra of the prepared samples. (**b**) Spectra subtracted from P25 spectrum. (**c**) Values of the bandgap energies estimated by the DPR method. (**d**) Colorimetric parameters of the samples in the CIE L*a*b* system. Inset shows photographs of the samples.

**Figure 5 nanomaterials-12-01501-f005:**
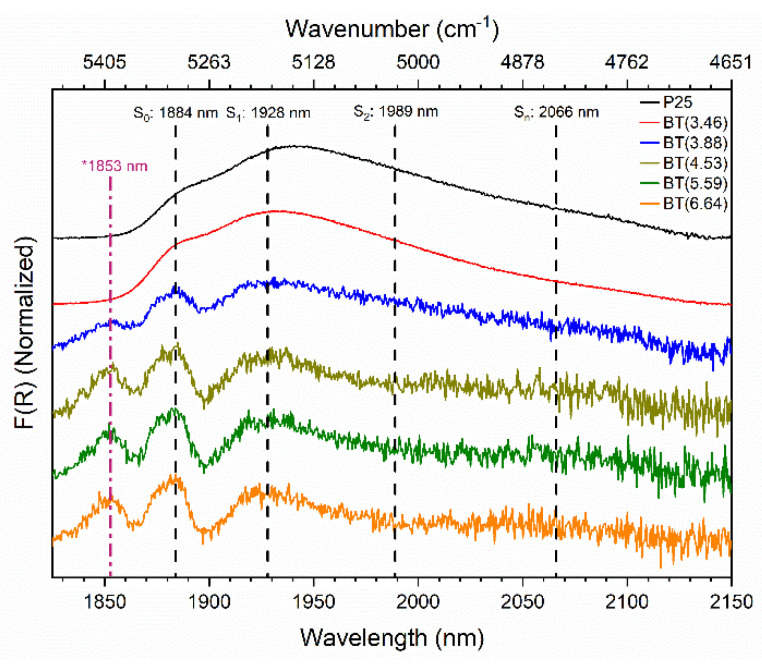
Normalized near-infrared spectra of blue titania samples in the region that provides information about the interaction of the partially reduced TiO_2_ surface with physisorbed water molecules.

**Figure 6 nanomaterials-12-01501-f006:**
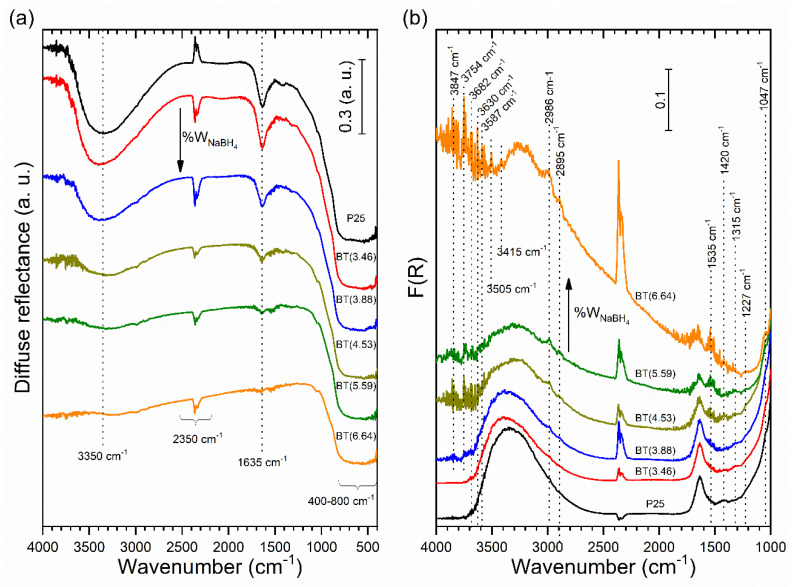
Infrared spectra of the prepared samples: (**a**) diffuse reflectance and (**b**) diffuse reflectance region transformed into Kubelka–Munk function. The ordinate axis of all spectra was shifted upwards for clarity.

**Figure 7 nanomaterials-12-01501-f007:**
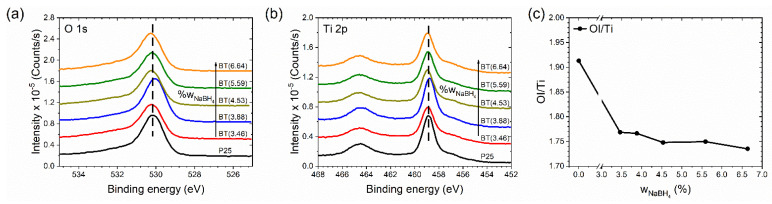
(**a**) Oxygen 1s and (**b**) titanium 2p spectra of the samples, where vertical dashed lines indicate the maximum peak for the P25 sample. (**c**) Structural oxygen/titanium ratio (OI/Ti) of the samples.

**Figure 8 nanomaterials-12-01501-f008:**
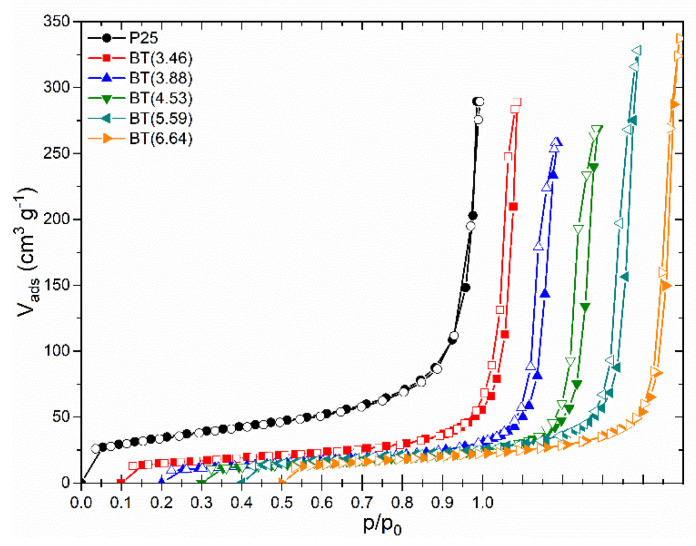
Experimental nitrogen adsorption/desorption isotherms of blue titania samples in comparison to P25. Isotherms were shifted by 0.1 along the p/p_0_ axis for clarity.

**Figure 9 nanomaterials-12-01501-f009:**
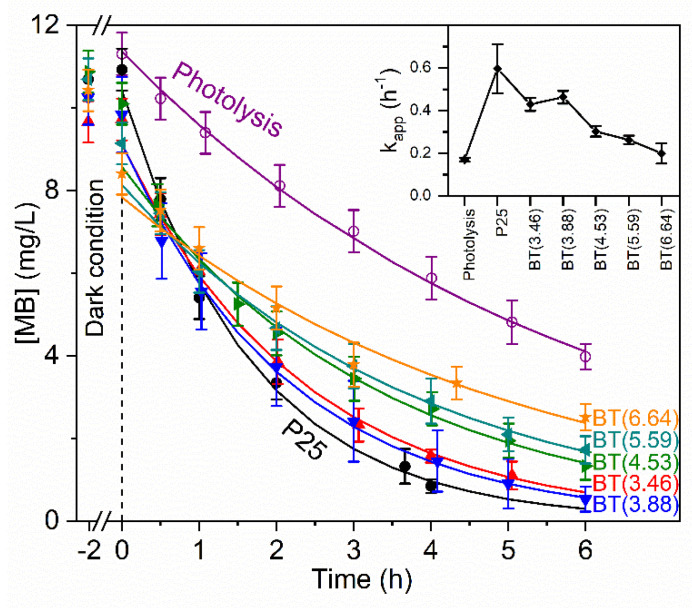
Kinetic curves of photolysis and photocatalytic degradation of methylene blue. Experimental measurements are in symbols and adjustments to kinetic models are in lines. (Inset) Values of the apparent rate constant (k_app_).

**Table 1 nanomaterials-12-01501-t001:** Prepared samples of blue titania (BT).

$\mathrm{Sample}\;(w_{NaBH_{4}},\;\%)$
P25 *
BT(3.46)
BT(3.88)
BT(4.53)
BT(5.59)
BT(6.64)

* P25 is commercial TiO_2_ as it was received.

**Table 2 nanomaterials-12-01501-t002:** Anatase fraction in the samples calculated from DRX data.

Sample	Weight Fraction of Anatase (%)	
	Spurr & Myers	Rietveld
P25	83.9 (1)	82.7 (1)
BT(3.46)	83.7 (2)	82.3 (2)
BT(3.88)	82.9 (2)	82.2 (1)
BT(4.53)	84.4 (1)	83.7 (2)
BT(5.59)	85.1 (2)	83.1 (2)
BT(6.64)	86.7 (4)	85.2 (4)

## Data Availability

Some data presented in this study are openly available in: Sabinas, Sergio (2022), “Dataset for “Blue titania: the outcome of defects, crystalline-disordered core-shell structure, and hydrophilicity change””, Mendeley Data, V2, doi:10.17632/ggbktsx5xj.2.

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
