# Peer review of "Blue Titania: The Outcome of Defects, Crystalline-Disordered Core-Shell Structure, and Hydrophilicity Change"

_nanomaterials, 2022, doi:10.3390/nano12091501_

Round 1
Reviewer 1 Report
The authors reported the properties of TiO2 P25 reduced by NaBH4 at five concentrations. It explains the changes of the structural characteristics of TiO2 with the increase of the reducing agent. The optical properties of the TiO2 and the catalytic degradation of MB are also studied. A conclusion is reported that the surface hydrophilicity of the sample during reduction is decrease. I recommend the publication of this work after major revision.
There were some problems with the manuscript. It could be summarized as the following:
- The position of the graph is not aligned to the text section in the main text.
- The annotation description of Figure 1e does not correspond with the text.
- A layer ofamorphous material appear in BT(6.64). When describing Figure S1, why does the authors say that there is no change before and after reduction?
- The quantities obtained by the Spurr & Myers equation of BT(88) is lower than BT(3.46)in Table 2. Why?
- The bands in Figure 3a and 3b are red shifting but not blue shifting.
- The curve positions are different in FigureS8a and S8b. Such as BT(6.64) is midway between BT(53) and BT(5.59) in Figure S8a, but in Figure S8b BT(6.64) is at the top.
- Why is the initial concentration different when different samplesdegrade MB in Figure 9?
- Blue titania dioxide degradation of MB effect is not good. Why use a long description?
搜索
复制
Reviewer 2 Report
I recommend publishing this manuscript in Nanomaterials after considering the following revisions:
- Authors have to provide results of reusability of the prepared photoatalysts by recycling/repetition experiments.
- Authors have to provide the spectrum of the lamp used for photocatalytic experiments.
- Authors have to compare band gap of P25 with literature data.
- Please calculate parentage removal efficiency of methylene blue by adsorption, photolysis and photocatalytic degradation by investigated photocatalyst.
-Authors have to explain obtained results of band gaps of photocatalyst and its photocatalytic activity? Please compare with literature data.
